# Low-Temperature Adaptive Dual-Network MXene Nanocomposite Hydrogel as Flexible Wearable Strain Sensors

**DOI:** 10.3390/mi14081563

**Published:** 2023-08-06

**Authors:** Kai Chen, Wenzhong Lai, Wangchuan Xiao, Lumin Li, Shijun Huang, Xiufeng Xiao

**Affiliations:** 1School of Resources and Chemical Engineering, Sanming University, Sanming 365004, China; 2Fujian Provincial Key Laboratory of Advanced Materials Oriented Chemical Engineering, College of Chemistry and Materials Science, Fujian Normal University, Fuzhou 350007, China

**Keywords:** organic hydrogel, MXene, dual network, wearable sensor, anti freezing

## Abstract

Flexible electronic devices and conductive materials can be used as wearable sensors to detect human motions. However, the existing hydrogels generally have problems of weak tensile capacity, insufficient durability, and being easy to freeze at low temperatures, which greatly affect their application in the field of wearable devices. In this paper, glycerol was partially replaced by water as the solvent, agar was thermally dissolved to initiate acrylamide polymerization, and MXene was used as a conductive filler and initiator promoter to form the double network MXene-PAM/Agar organic hydrogel. The presence of MXene makes the hydrogel produce more conductive paths and enforces the hydrogel’s higher conductivity (1.02 S·m^−1^). The mechanical properties of hydrogels were enhanced by the double network structure, and the hydrogel had high stretchability (1300%). In addition, the hydrogel-based wearable strain sensor exhibited good sensitivity over a wide strain range (GF = 2.99, 0–200% strain). The strain sensor based on MXene-PAM/Agar hydrogel was capable of real-time monitoring of human movement signals such as fingers, wrists, arms, etc. and could maintain good working conditions even in cold environments (−26 °C). Hence, we are of the opinion that delving into this hydrogel holds the potential to broaden the scope of utilizing conductive hydrogels as flexible and wearable strain sensors, especially in chilly environments.

## 1. Introduction

With the improvement of medical care, wearable sensors are widely used in the fields of medical health monitoring, disease diagnosis, and human-computer interaction [1,2,3]. Sensors are devices that receive and respond to signals and stimuli from the environment. Sensors can be broadly classified into two categories: physical sensors and chemical sensors [4]. Physical sensors are devices that provide information about the physical properties of a system, such as photoelectric sensors and strain sensors. On the other hand, chemical sensors are devices that convert chemical information, ranging from the concentration of specific sample components to overall composition analysis, into analytically useful signals, such as humidity sensors and glucose sensors [5]. The sensors need to be able to capture the wearer’s movements or changes in environmental humidity, temperature, etc. and convert them into electrical signals [6,7,8]. And they should be able to adhere to the wearer’s skin, which requires good biocompatibility of the sensor substrate. Hydrogels are ideal for wearable sensors because of their good flexibility, biocompatibility, and controllable electrical conductivity [9,10]. Generally speaking, the regulation of the conductivity of hydrogels can be achieved in the following four ways: the introduction of ionic salts, the direct gelation of conductive materials, the combination of hydrogel precursors and conductive fillers, and the in situ polymerization of conductive polymers in precursor hydrogels. Due to the introduction of conductive material, more conductive paths are formed inside the hydrogel, which endows the hydrogel with good electrical conductivity [11].

Agar is a polysaccharide component widely distributed in seaweed that has the advantages of abundant source, low price, safety, and non-toxicity [12]. Due to its high melting point and high gelation temperature, it can be used as an ideal carrier material for sterilization and the provision of nutrients [13,14]. And because it retains its original helical structure, it can undergo sol-gel transformation after repeated heating and cooling. However, the poor mechanical properties of Agar limit its application in hydrogels. Therefore, the synthesis of functional Agar composite hydrogels with excellent mechanical properties by combining them with other conductive nanomaterials has become a hot research topic [15,16,17]. MXene is a two-dimensional structure of carbonized (nitride) transition metal materials, and because of its good hydrophilic, conductive, and adjustable properties, it is often added to the hydrogel as a conductive filler [18,19,20]. And its rich surface functional groups, activated in the hydrogel process not only as a filler but also as a crosslinker, initiator, and multifunctional nano-filler, enhance the mechanical properties of hydrogel. In summary, MXene is a new nanomaterial with broad application prospects [21,22].

Herein, in order to solve the problem of poor mechanical and electrical properties of Agar, we prepared a double-network nanocomposite hydrogel named MXene-PAM/Agar. The double network structure was constructed by introducing polyacrylamide into the agar network, which endowed the hydrogel with excellent mechanical properties. In addition, the mechanical and electrical properties of the hydrogel were further enhanced by adding MXene nanosheets to the hydrogel. The acrylamide, agar, and MXene nanoplates were added to the glycerin–water solution in a one-pot process. The acrylamide monomers were polymerized to form long chains and cross-linked under Mxene initiation [23]. The agar was dissolved at high temperature to form long chains, and after cooling, it was cross-linked to form another layer of network structure, thus forming double-network MXene-PAM/Agar organic hydrogels. In addition, the glycerin–water solution replaced the deionized water solvent, which made heat difficult to disperse and promoted the formation of hydrogel [24]. The hydrogen bond between glycerin and water could hinder the formation of ice crystals at low temperatures, but the cold resistance of the hydrogel was improved [25,26]. The hydrogel can withstand more than 80% compressive strain and 0.35 MPa compressive stress and 1000% tensile strain and 0.25 MPa tensile stress. The experimental results showed that the hydrogel-based strain sensor could maintain good flexibility, extensibility, and conductivity in a cold environment (−25 °C), thus effectively improving the long-term stability of the hydrogel. Therefore, the study of the hydrogel further extended the application scenarios of conductive hydrogel as flexible wearable strain sensors, especially in extreme environments.

## 2. Experimental Section

### 2.1. Materials

Glycerol was purchased from Shanghai Test National Pharmaceutical Group Chemical Reagent Co., Ltd. (Shanghai, China). N-N′methylene bisacrylamide (MBAA) and agar were purchased from Shanghai Aladdin Biochemistry Co., Ltd. (Shanghai, China). Aluminum titanium carbide (MAX), hydrofluoric acid, ammonium persulfate (APS), acrylamide (AM), and tetramethyl ethylenediamine (TEMED) were purchased from Shanghai Macklin Biochemical Technology Co. (Shanghai, China). Deionized water was used for all experiments.

### 2.2. Preparation of MXene Nanosheets

MXene was prepared by etching Ti_3_AlC_2_ with HF (Figure 1). Firstly, a moderate magnet was put into a PTFE reaction kettle, and 40 mL of hydrofluoric acid was transferred into the reaction kettle. Secondly, 1 g of MAX was slowly added to the hydrofluoric acid to avoid drastic changes due to the exothermic reaction and stirred at room temperature for 24 h. After the reaction was complete, the mixed solution was washed several times by centrifugation with deionized water at a centrifugation rate of 10,000 rpm until a pH value of about 6.5 was reached, followed by cryogenic ultrasound for 1 h. The sample was then completely frozen and transferred to −80 °C for freeze-drying, and then the MXene nanosheets were prepared.

### 2.3. Preparation of MXene-PAM/Agar Organic Hydrogel

Firstly, acrylamide monomer with a concentration of 15% was dissolved in DI water, and then agar powder with a concentration of 2% was dissolved in AM solution by heating at 95 °C for 10 min to form an evenly distributed solution. Then, MXene suspension with different concentrations, APS initiator, and crosslinking agent MBAA were added to the mixed solution, stirred violently, quickly poured into Teflon molds, and left for 30 min to form MXene-PAM/Agar hydrogel. Similarly, MXene-PAM-Agar organic hydrogels were prepared by substituting a glycerol–water solution for deionized water.

### 2.4. Characterization

The physical phase composition of MAX and MXene was analyzed by X-ray diffraction (XRD) (Japanese Ultima IV) using a Cu target and a working voltage range of 20–60 kV. The sample was placed horizontally and scanned with a measuring range of 10–80° and a speed of 20° min^−1^. The microscopic morphology of MAX, MXene powders, and MXene-PAM/Agar organic hydrogels was analyzed by scanning electron microscope (SEM) (Regulus 8100). Before the SEM, the hydrogel was frozen by liquid nitrogen, and the water was removed by freeze-drying. The hydrogel samples were placed onto copper studs and then coated with a thin layer of gold/palladium using sputter deposition for a duration of 60 s.

### 2.5. Mechanical Tests of MXene-PAM/Agar Organic Hydrogel

The mechanical properties of the hydrogels were determined using a universal material testing machine (WDW-05, Si Pai Inc. Shanghai, China) at room temperature. In the compression test, a cylindrical hydrogel with a 15 mm diameter and 10 mm height was placed on the lower plate and compressed at a strain rate of 10 mm min^−1^. In tensile tests, hydrogels were made into cylindrical samples (length 50 mm, diameter 3 mm) and stretched with a strain rate of 50 mm min^−1^. The modulus of elasticity was determined by calculating the slope of the linear region of the stress–strain curve. The toughness was calculated by integrating the area under the stress–strain curve.

### 2.6. Electrical Measurement of MXene-PAM/Agar Organic Hydrogel

The resistivity of the MXene-PAM/Agar organic hydrogel with different concentrations of MXene was tested by a digital four-probe tester (Suzhou Crystal Lattice, Suzhou, China). The resistance variation of MXene-PAM/Agar organic hydrogel with different deformations was measured by a LCR meter (TH2832, Changzhou, China). Each end of the hydrogel was inserted into a copper wire that connects to the TH LCR meter and further encapsulated with two medical PU tapes to minimize environmental interference. The MXene-PAM/Agar organic hydrogel-based strain sensor was connected to the TH LCR meter to record the resistance changes in time.

### 2.7. The DSC Testing of MXene-PAM/Agar Hydrogels

Differential scanning calorimetry (DSC) was used to study the variation of heat flow rate with temperature in samples and reference samples by commanding the changes in temperature. Samples were encapsulated in a sealed aluminum crucible for testing, and an empty disc was used as an inert reference. Experimental data were recorded at a rate of 1 Hz under a nitrogen flow rate of 50 μL/min. The measurement range was −70 °C to 25 °C, and the temperature changed at a rate of 5 °C/min.

### 2.8. Fabrication of MXene-PAM/Agar Organic Hydrogels Based Sensors

To fabricate the strain sensors, MXene-PAM/Agar hydrogels were cut into strips with a size of 25 mm × 10 mm (1 mm thickness). Each end of the hydrogel was inserted into a copper wire that connects to the TH LCR meter and further encapsulated with two medical PU tapes to minimize environmental interference. The MXene-PAM/Agar organic hydrogel-based strain sensor was connected to the TH LCR meter to record the resistance changes in real time. When monitoring human motion, the sensor was mounted directly on the volunteer’s skin, and the electrical signal was recorded in real time by the LCR tester.

## 3. Results and Discussion

### 3.1. Design Principles and Material Synthesis

In this study, the MXene-PAM/Agar organic hydrogel was prepared by a one-pot method (Figure 2). Firstly, the agar was added to the acrylamide solution and heated to disperse the agarose macromolecules in the solution. The ultrasound-dispersed MXene suspension was mixed with an acrylamide agar solution, and then the APS initiator and crosslinker MBAA were added. The first polyacrylamide network was formed by thermally initiated cross-linking under the action of an acrylamide crosslinker. After cooling, the agarose macromolecules formed a double helix structure to construct the second network, and the MXene-PAM/Agar organic hydrogel with a double network was obtained. In this system, the polyacrylamide network helped the hydrogel maintain good strength and toughness. The second network of agarose double helix structure had a large number of hydrogen bonds, which could be broken and re-formed as a sacrifice crosslinking to dissipate energy when subjected to external forces, endowing the hydrogel with good stretchable properties [27,28]. In addition, the addition of MXene could make the system form more hydrogen bonds to dissipate energy, which would further improve the mechanical properties of hydrogels. At the same time, the unique electrical properties of MXene gave the hydrogel excellent conductivity [29]. In addition, the use of a glycerin-water solvent instead of water could not only retain heat and make the heat initiation of gelation faster but also endow the hydrogel with good frost resistance [30,31]. These excellent properties of MXene-PAM/Agar organic hydrogel make it an excellent material for flexible wearable strain sensors, especially since it has the potential to work at low temperatures.

### 3.2. Characterization of MXene Nanosheets

The X-ray diffraction (XRD) patterns of the raw materials MAX and MXene nanosheets are presented in Figure 3a. The diffraction peaks observed in the XRD analysis of the raw material MAX can be accurately matched with the hexagonal crystal structure of Ti_3_AlC_2_ (JCPDS No. 52-0875) [32]. After LiF and HCl etching, intercalation, and stripping, the peak (104) of MAX disappeared, the characteristic peak (002) shifted to 7.1, and the peak value decreased [32,33]. This was This was due to the etching between the MAX layer and the aluminum layer, embedding the surface active groups Tx(such as -OH, -F, and -O), thus increasing the layer spacing [34]. Furthermore, the presence of a diminishing peak (110) at 61.5° indicated that the crystallinity and degree of order of TiAlC_2_ decreased as the reaction progressed [35]. The morphology of Max and MXene nanosheets was observed by scanning electron microscopy (SEM). As shown in Figure 3b, and 3c, the original MAX was a solid structure with no separation between the layers; the resulting etched MXene exhibited a distinctive accordion-like multilayer nano sheet structure, characterized by diameters that varied between 0.5 μm and 5 μm. The above XRD and SEM results indicated that the MXene was successfully prepared. As shown in the SEM image of the MXene-PAM/Agar hydrogel, the hydrogel formed a three-dimensional network structure (Figure 3d). EDS analysis of the hydrogel showed that the distribution of Ti elements in the hydrogel was relatively uniform, indicating that MXene was evenly doped into the hydrogels.

### 3.3. Mechanical Properties of MXene-PAM/Agar Organic Hydrogel

The obtained MXene-PAM/Agar organic hydrogel with a double network structure exhibited excellent mechanical properties. As shown in Figure 4a, the hydrogel showed good stretchable properties and could be stretched to 400% of its original state. The hydrogel that can dissipate external stress has no scratches or gaps on its surface after being cut by a knife (Figure 4b). Moreover, the hydrogel could endure large compressive deformation and recover rapidly to its original shape after removing the external stress, which showed the excellent mechanical resilience of this hydrogel (Figure 4c). To further assess the mechanical properties of the hydrogels, a comprehensive set of tensile and compression tests was performed (Figure 4d,e). The double network MXene-PAM/Agar hydrogel included a rigid and brittle first network and a soft and flexible second network. The toughening theory of double-network structure was mainly based on the “sacrificial bond theory” [36,37]. After an external force was applied to the hydrogel, the first network within the structure was disconnected, effectively dissipating energy and providing protection for the second network. This allowed the second network to maintain pressure and store elasticity, thereby increasing the overall strength of the hydrogel. The MXene-PAM/Agar organic hydrogel has a double helix structure and a large number of hydrogen bonds. When the hydrogel was stretched or compressed, the hydrogen bond fracture was accompanied by a large amount of energy dissipation and dispersion stress, so the hydrogel was not broken after being stretched or compressed. When the external force is removed, hydrogen bond recombination restores the original state.

Herein, we focused on the effect of different MXene additions (0–1.0 wt%) on the mechanical properties of the MXene-PAM/Agar hydrogels. Five groups of hydrogels with different MXene content (0, 0.4 wt%, 0.6 wt%, 0.8 wt%, and 1.0 wt%) were prepared and labeled as M-0, M-1, M-2, M-3, and M-4, respectively. As shown in Figure 5a, the tensile strain-stress curves showed that when the concentration of MXene increases from 0% to 1%, the fracture strain increases from 612% to 1310%, which is twice that of PAM/Agar hydrogel. The corresponding Young’s modulus (Figure 5b) and the tensile toughness (Figure 5c) of hydrogels were obtained from the tensile strain-stress curves of hydrogels with different MXene concentrations. From the figure, it could be concluded that the toughness and Young’s modulus of the hydrogels were improved with the addition of MXene nanosheets. The enhanced mechanical strength could be attributed to the abundance of functional groups (-OH, -O, etc.) present on the MXene surface. These functional groups had the ability to form hydrogen bonds with the PVA chains, which further strengthened the hydrogel and improved its overall mechanical properties. Importantly, it should be noted that the Young’s moduli of all the hydrogels remained below 10 kPa, ensuring that they retained a desirable level of softness similar to that of human skin. Similarly, the addition of MXene nanosheets also improved the compression strength of the hydrogels (Figure 5d). All five groups of hydrogels could be compressed to 80% without rupture, indicating that the hydrogels had good compressibility. The compressive modulus (Figure 5e) and compressive toughness (Figure 5f) have the same variation trend as the tensile modulus and tensile toughness. In short, the MXene-PAM/Agar organic hydrogels prepared by MXene have good tensile properties, mechanical strength, and toughness.

### 3.4. Freezing Resistance of MXene-PAM/Agar Organic Hydrogel

The hydrogel with high water content freezes at low temperatures, and its conductivity decreases obviously, so it cannot work as a sensor below 0 °C, which limits the application of hydrogel in flexible sensors [38]. The replacement of the solvent in the hydrogel with a glycero–water solvent solves this problem, and the formation of hydrogen bonds between glycerol and water inhibits the formation of ice crystals and enhances the anti-freezing performance of the hydrogel. According to the results of Differential scanning calorimetry (DSC), the anti-freezing temperature of the organic hydrogel with glycerol added reached −39.7 °C, and the origin hydrogel was −29.1 °C (Figure 6a). The hydrogel was stretched at low temperatures and then ruptured, forming ice crystals that were unable to work as strain sensors (Figure 6b). While the organic hydrogel could still be stretched and twisted without fracture at low temperatures (Figure 6c) and successfully lit a light bulb. The organic hydrogel showed freeze-tolerance and potential when applied below 0 °C. Furthermore, the conductivity of MXene-PAM/Agar hydrogel and MXene-PAM/Agar organic hydrogel was measured as a function of temperature (Figure 6d). The results showed that the hydrogels affected by temperature and conductivity were rising at room temperature and becoming unstable, and the MXene-PAM/Agar organic hydrogel during this process was stable almost all the way up to 0.8 S·m^−1^. The MXene-PAM/Agar organic hydrogel was assembled into a strain sensor, which was then affixed to the finger and monitored for finger bending in cold environments, and the hydrogel sensor showed good sensitivity.

### 3.5. Strain Sensing Performance of the MXene-PAM/Agar Organic Hydrogel

The electrical conductivity of hydrogels can convert external stimuli (such as stress, temperature, pH, moisture, light intensity, etc.) into detectable electrical signals [39]. Due to the incorporation of MXene powder, MXene-PAM/agar hydrogels exhibited electrical conductivity, and the effects of different additions (0, 0.4 wt%, 0.6 wt%, 0.8 wt%, and 1 wt%) of MXene on the electrical conductivity of the organic hydrogels were further investigated. The electrical conductivity of five groups of composite organic hydrogels was tested with a four-probe resistivity tester. As shown in Figure 7a, the conductivity was higher (1.02 S·m^−1^) when the MXene content was 0.6 wt%, whereas it decreased when the MXene content continued to increase. This might be due to the formation of an effective conductive pathway when the MXene content was moderate, while the stacked conductive pathway of MXene decreased the conductivity of hydrogels when the MXene content continued to increase. The organic hydrogel with a MXene content of 0.6 wt% was selected for the follow-up study.

The sensitivity of the strain sensor is quantified by its gauge factor (GF) and calculated by the equation: GF = (ΔR/R_0_)/ε, where ΔR = R−R_0_, R and R_0_ are the original resistance and the resistance under a certain deformation, respectively, ε is the corresponding strain [40]. The strain sensing performance of the MXene-PAM/Agar organic hydrogel was evaluated via the brightness of the LED bulb at 0%, 100%, and 200% strain (Figure 7b). It could be observed that the bulb brightness gradually decreased as the hydrogel deformation increased. As shown in Figure 7c, the change in resistance for five stretching–relaxation cycles performed at different strains (25%, 50%, 100%, and 200%) can be clearly observed. ΔR/R_0_ increased monotonically with the deformation variable, and the strain was similar for each cycle. This indicated that the MXene-PAM/Agar organic hydrogel could monitor the resistance change caused by deformation in a certain range and had good reliability. And there was a linear relationship between the ΔR/R_0_ and the strain of 0–200%, with GF = 1.66 at the 0–50% strain and GF =2.99 at the 50%–200% strain (Figure 7d). In comparison to the majority of previously reported flexible strain sensors, the MXene-PAM/Agar organic hydrogel sensor exhibited superior sensitivity and stability across a broad range of strain variations. Additionally, we conducted evaluations to assess the response time and recovery time of the hydrogel strain sensor. The response time and recovery time of the sensor under rapid strain were 150 ms and 200 ms, indicating that the hydrogel sensor has a faster response rate than the hydrogel-based sensor (Figure 7e). Compared with most reported hydrogel-based strain sensors, it has faster response and stability over the strain range. The long-term stability of the hydrogel sensor was further tested. At a strain of 200%, the strain sensor shows a stable response signal, i.e., no fluctuation in amplitude or waveform, during the load-to-unload cycles for 1000 consecutive seconds (Figure 7f), confirming the high stability and reliability of the device.

### 3.6. The MXene-PAM/Agar Hydrogel Based Strain Sensor for Real-Time Monitoring of Human Movements

The strain sensor based on MXene-PAM/Agar organic hydrogel had the characteristics of high sensitivity, a wide sensing range, a fast response time, and good stability, which were expected to be applied in wearable devices to detect human movements [41]. We constructed a dual VHB sandwich hydrogel strain sensor that increases hydrogel stability and reduces water loss from human contact. The sensor can be directly adhered to human skin and can detect human motions such as fingers, elbows, knees, and wrists. As shown in Figure 8a, the ΔR/R_0_ changes from 0 to 105.5 as the finger bends gradually from the straight state to 90°. The human motion electrical signal ΔR/R_0_ increased from 0 to 101% with five repetitions of bending relaxation, with basically the same change. Similar changes were observed in knee, elbow, and wrist flexion (Figure 8b–d). We further attached the wearable strain sensor to the arm to detect muscle movements. As the arm gradually bends, the biceps gradually swell, and the ΔR/R_0_ increases from 0% to 248% (Figure 8e). Therefore, the MXene-PAM/Agar hydrogel strain sensor can monitor muscle movement during exercise. In addition, based on the hydrogel’s low mechanical lag, good compressive resistance, electrical conductivity, resilience, and stability, we sandwiched the hydrogel column between the sheet electrodes and placed it directly under the heel to detect the load on the foot during movement. As shown in Figure 8f, a steady repetitive response was observed during the five-cycle lift-and-drop cycle. The hydrogel-based sensor had a unique design and could be used to measure plantar pressure. The MXene-PAM/Agar organic hydrogel sensor holds significant potential for various applications, including sports injury prevention and sports biomechanics. Therefore, the hydrogel sensor can accurately respond to the full range of human movements and has good repeatability and stable signal output.

## 4. Conclusions

In summary, the MXene-PAM/Agar hydrogel was successfully fabricated by adding MXene nanosheets into a hot AM-Agar mixed solution, thermally triggering AM crosslinking, and forming the double network MXene-PAM/Agar hydrogel after cooling. In addition, glycerol was added to the hydrogel system to replace part of the water, which made the hydrogel have good frost resistance. The MXene-PAM/Agar hydrogel has better mechanical properties and frost resistance, which could be used as a wearable strain sensor with a satisfactory sensitivity factor and stability. The Mxene-PAM/Agar hydrogel-based sensor can be applied to detect various physiological activities of the human body, such as finger flexion, elbow flexion, etc., showing the great potential of its application for sports biomechanics, human motion monitoring, and medical examination.

## Figures and Tables

**Figure 1 micromachines-14-01563-f001:**
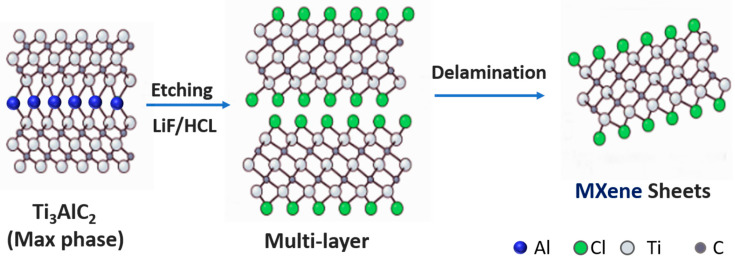
Schematic of the synthesis of MXene nanosheets.

**Figure 2 micromachines-14-01563-f002:**
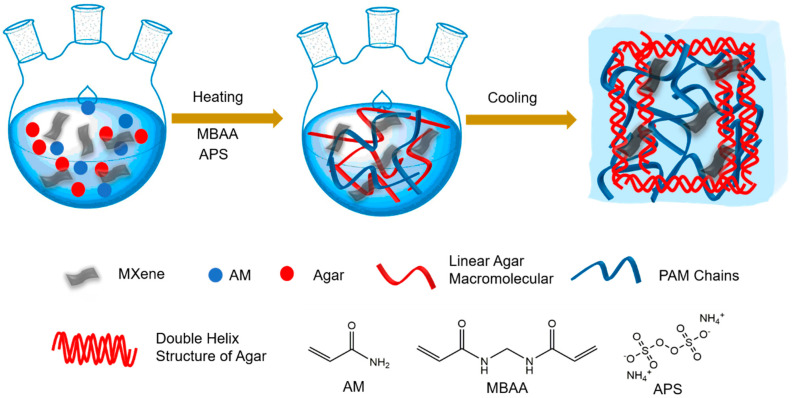
Schematic diagram of fabrication process of the MXene-PAM/agar organic hydrogel.

**Figure 3 micromachines-14-01563-f003:**
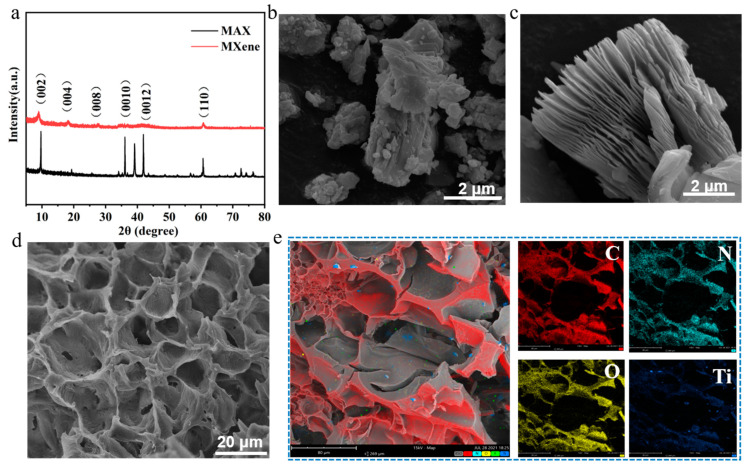
Characterization of MXene powder and MXene-PAM/Agar hydrogel. (**a**) XRD images of MAX and MXene. SEM images of MAX (**b**) and MXene (**c**). (**d**) SEM images of the MXene-PAM/Agar. (**e**) EDS analysis images of the MXene-PAM/Agar hydrogel.

**Figure 4 micromachines-14-01563-f004:**
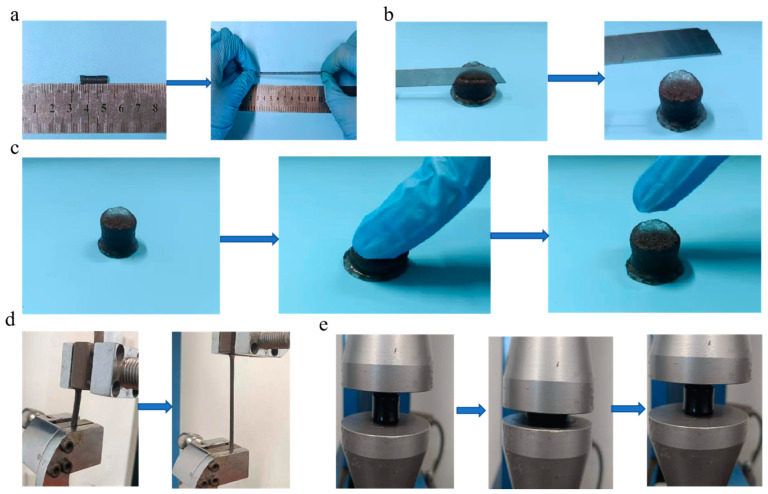
Schematic diagram of the mechanical performance of MXene-PAM/Agar organic hydrogel (0.6 wt% MXene). (**a**) MXene-PAM/Agar organic hydrogel was stretched. (**b**) cut with a knife; (**c**) hydrogel rebounded after finger pressure removed. (**d**) The organic hydrogel was stretched by universal materials tester (**e**) and compressed by universal materials tester, then recovered after clamp lifting.

**Figure 5 micromachines-14-01563-f005:**
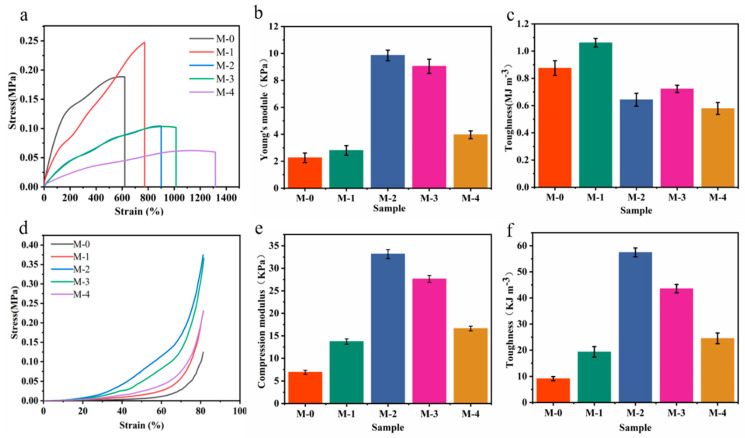
Mechanical properties of MXene-PAM/Agar organic hydrogels. The tensile strain-stress curves (**a**), Young’s modulus (**b**), and tensile toughness (**c**) of the MXene-PAM/Agar organic hydrogels. The compressive strain-stress curves (**d**), compressive modulus (**e**), and compressive toughness (**f**) of the hydrogels with different MXene contents.

**Figure 6 micromachines-14-01563-f006:**
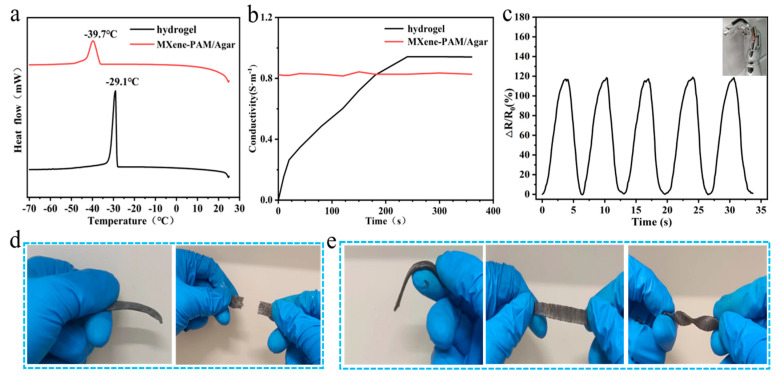
Comparison of freezing resistance of the hydrogel and organic hydrogel. (**a**) the DSC curves of hydrogel and MXene-PAM/Agar organic hydrogel showing their crystallization points; (**b**) Conductivity as a function of time after transferring hydrogel and organic hydrogel from a −26 °C refrigerator to room temperature; (**c**) organic hydrogel worked as a wearable strain sensor at −26 °C. (**d**) The hydrogel fractured after being stretched at low temperature. (**e**) The MXene-PAM/Agar organic hydrogel was stretched and twisted without fracture at low temperatures.

**Figure 7 micromachines-14-01563-f007:**
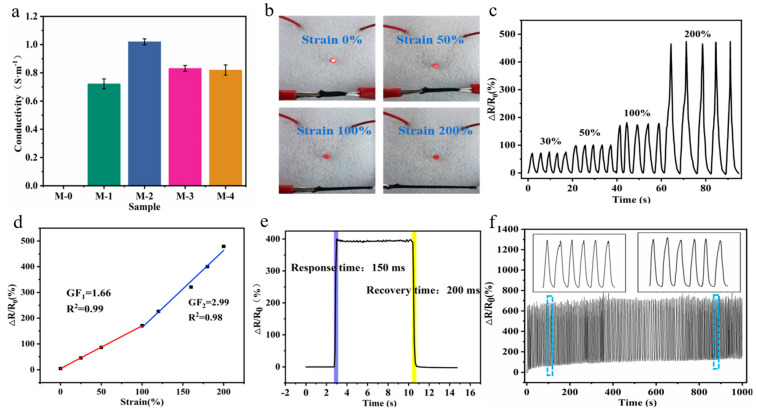
Electromechanical performances of the MXene-PAM/Agar organic hydrogel were applied as flexible strain sensors. (**a**) Conductivity of MXene−PAM/Agar organic hydrogels with different MXene contents. (**b**) When the MXene-PAM/Agar organic hydrogel was subjected to different tensile strains (0%, 50%, 100%, and 200%), the LED beads exhibited varying levels of brightness. (**c**) Variation of ∆R/R_0_ of the organic hydrogel at 30%, 50%, 100%, and 200%. (**d**) The relative resistance of the hydrogel sensor changed as the tension strain ranged from 0% to 200%. The GF could be determined by calculating the slope of the fitted regression line. (**e**) The response time and recovery time of the hydrogel sensor. (**f**) The change in resistance of the hydrogel sensor was measured by cycling for 1000 s at a 200% strain.

**Figure 8 micromachines-14-01563-f008:**
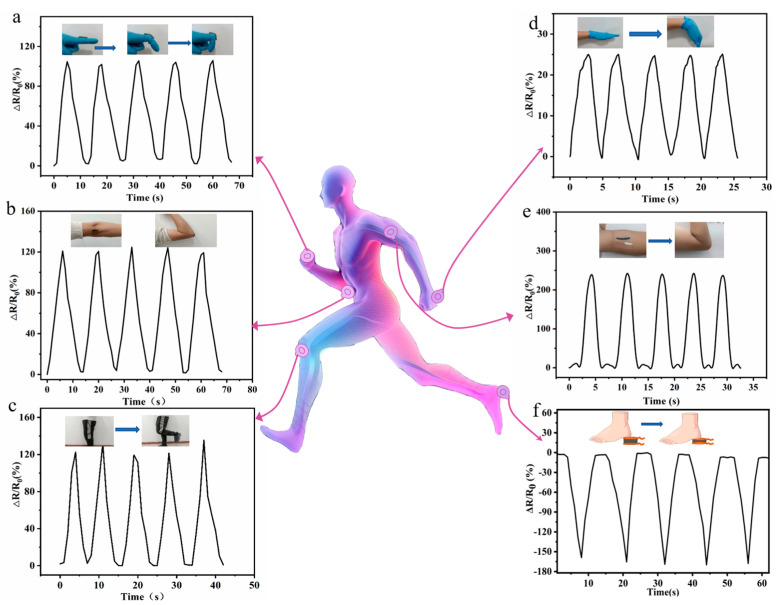
Real-time monitoring of human motions with finger (**a**), elbow (**b**), knee (**c**), wrist (**d**), and arm muscle movements (**e**). (**f**) The real-time monitoring of foot planter pressure during locomotion.

## Data Availability

The datasets generated during and/or analyzed during the current study are available from the corresponding author upon reasonable request.

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
