# Peer review of "Low-Temperature Adaptive Dual-Network MXene Nanocomposite Hydrogel as Flexible Wearable Strain Sensors"

_micromachines, 2023, doi:10.3390/mi14081563_

Round 1

Reviewer 1 Report

see my comments

1.      Be consistent in unit denotations (see 18 and 20; % denotation-space to the figure)

2.      Cite very recent works on hydrogels such as: https://pubs.rsc.org/en/content/articlehtml/2022/tc/d2tc03102g and https://www.mdpi.com/2310-2861/9/3/250

3.      Stretchability against electrical conductivity must be tested in more detailed manner

4.      How the additions of aluminium layer increased spacing, needs referencing

Reviewer 2 Report

Comments to Author

The authors should present some brief background information in the introduction section about the various types of sensor models working to detect water and glucose. Since this will establish a direct relation with the objective of this article.

In section 2 of the article, the authors have appropriately explained the procedure for the preparation of various materials but the chemical modeling part seems missing. The fabrication process can be explained with more effectiveness using chemical notations.

Section 3 of the article seems complete to me, with no further modification required, The manuscript contains some typo errors.

Increase the font size of the legends in the figure to the appropriate dimension to make them easy to read. None of the statistics has readable legend sizes, see Figures 6, 7 etc.

The article should be read thoroughly. Check the grammar. 

 Minor editing of English language required

Reviewer 3 Report

Chen et al. successfully developed the MXene-PAM/Agar hydrogel through the addition of MXene nanosheets to a hot AM-Agar mixed solution. The AM crosslinking was thermally triggered, resulting in the formation of a double network MXene-PAM/Agar hydrogel upon cooling. To improve frost resistance, glycerol was incorporated into the hydrogel system, replacing a portion of water. The MXene-PAM/Agar hydrogel demonstrated enhanced mechanical properties and frost resistance, making it suitable for use as a wearable strain sensor with excellent sensitivity and stability. This hydrogel-based sensor, with its satisfactory sensitivity factor and stability, holds promise for detecting various physiological activities of the human body, such as finger and elbow flexion. Its potential applications span sports biomechanics, human motion monitoring, and medical examinations. The research presented in this study is highly relevant to readers of Micromachines. However, before publication, the authors should address certain key points to bolster the credibility and robustness of their research.

1.      The stability of the MXene-PAM/Agar hydrogel is a critical challenge, especially during long-term use or under extreme environmental conditions. It is necessary to explore methods to enhance its stability.

2.      The sensitivity and dynamic range of the MXene-PAM/Agar hydrogel sensor are essential indicators for strain detection. Further optimization is required to meet various application demands.

3.      Scaling up the technology for large-scale production poses a challenge that involves addressing issues related to the scalability of the fabrication process.

4.      Comfort and reliability are crucial factors for flexible wearable strain sensors, as they play a vital role in ensuring long-term wearability and dependable detection. A comprehensive consideration of material and design factors is necessary.

Moderate editing of English language required

Round 2

Reviewer 1 Report

well done